# Obesity prevention in the early years: A mapping study of national policies in England from a behavioural science perspective

Helen Croker[1]*, Simon J. Russell[1], Aswathikutty Gireesh[1], Aida Bonham[1], Corinna Hawkes[2], Helen Bedford[1], Susan Michie[3], Russell M. Viner[1]

**1** Population, Policy and Practice Research and Teaching Department, UCL Great Ormond Street Institute of Child Health, London, United Kingdom, **2** School of Health Sciences, Division of Health Services Research & Management City, University of London, Northampton Square, London, United Kingdom, **3** UCL Centre for Behaviour Change, Department of Clinical, Educational and Health Psychology, University College London, London, United Kingdom

* h.croker@ucl.ac.uk

## Abstract

### Background

Evidence indicates that early life is critical for determining future obesity risk. A sharper policy focus on pregnancy and early childhood could help improve obesity prevention efforts. This study aimed to systematically identify and categorise policy levers used in England with potential to influence early life course (pregnancy, 0–5 years) and identify how these interface with energy balance behaviours. The objective is to identify gaps and where further policy actions could most effectively focus.

### Methods

A behavioural science approach was taken using the Capability-Opportunity-Motivation-Behaviour (COM-B) model and Behaviour Change Wheel (BCW) framework. The key determinants of energy balance in the early years were identified from the Foresight Systems Map. Policy actions were scoped systematically from available literature, including any health or non-health policies which could impact on energy balance behaviours. Foresight variables and policy actions were considered in terms of COM-B and the BCW to determine approaches likely to be effective for obesity prevention and treatment. Existing policies were overlaid across the map of key risk factors to identify gaps in obesity prevention and treatment provision.

### Results

A wide range of policy actions were identified (n = 115) to address obesity-relevant risk factors. These were most commonly educational or guidelines relating to environmental restructuring (i.e. changing the physical or social context). Scope for strengthening policies relating to the food system (e.g. the market price of food) and psychological factors contributing to obesity were identified. Policies acted via all aspects of the COM-B model, but there

**Data Availability Statement:** All relevant data are within the manuscript and its Supporting Information files.

**Funding:** This report is independent research funded by the National Institute for Health Research (NIHR) Policy Research Programme (Policy Research Unit: Obesity/ PR-PRU- 0916-21001). The views expressed in this publication are those of the authors and not necessarily those of the NIHR or the Department of Health and Social Care. Grant awarded to RMV The funders had no role in study design, data collection and analysis, decision to publish, or preparation of the manuscript.

**Competing interests:** The authors have declared that no competing interests exist.

was scope for improving policies to increase capability through skills acquisition and both reflective and automatic motivation.

## Conclusions

There is substantial policy activity to address early years obesity but much is focused on education. Scope exists to strengthen actions relating to upstream policies which act on food systems and those targeting psychological factors contributing to obesity risk.

## Introduction

Rates of overweight and obesity in children have increased worldwide since the 1980s, and despite a slowing recently, it remains an important public health issue with serious consequences for health [1, 2]. In the UK, approximately 10% of children aged 4–5 years and 20% of those aged 10–11 years were living with obesity in 2018/19 [3]. Children in the UK are also developing obesity at a younger age and therefore are likely to accumulate greater levels of overweight across their lifetimes [4]. Behaviour-changing interventions to treat children with overweight and obesity can reduce body mass index (BMI) but effects tend to be small [5] and effective prevention programmes have generally proved elusive [6]. There is growing evidence for the importance of pregnancy and maternal factors, including maternal weight, general health and the environment, for programming weight gain in offspring across the life-course [7, 8]. Pre and postnatal growth, and their interaction, are important for establishing risk factors for obesity [7]. Children from disadvantaged backgrounds have higher rates of obesity and, in England, these inequalities have increased over the past decade [9]. Early life is likely to be critical; a recent systematic review showed consistent associations between rapid growth in the first two years of life and later obesity, with the strongest effects in children of lower socio-economic status (SES) and ethnic minority groups [10]. Evidence also suggests that children of lower SES have greater exposure to several early life risk factors for obesity, including pre-pregnancy maternal obesity and diabetes, low birth weight and poorer early life nutrition [11]. This suggests that a sharper policy focus on pregnancy and early childhood could be useful for improving the effectiveness of obesity prevention efforts and for addressing health inequalities.

The complex nature of obesity, with its numerous interacting contributing factors, requires a comprehensive, coordinated and sustained policy response [12]. Policy levers have been described as "instruments that can be adjusted by governments to achieve system-wide change" [13]. Adjusting or applying a policy lever should significantly impact on the system, and in this case influence behaviours or determinants of behaviour related to child obesity [14]. Policy actions are available through the 'layers of influence on health', from the individual, through family, social and community networks ('downstream' influences), to wider socioeconomic and environmental conditions and up to national level ('upstream influences') [15]. Therefore, there are a range of periods and settings where policy levers may potentially be applied during the early life course. National obesity policy has tended to focus on individual responsibility and locally-led actions [16–19]. However, a comprehensive mapping of existing policies in England to intervene during the critical early years part of the life course, and whether they target appropriate behaviours or behavioural influences has not been conducted to the authors' knowledge. An improved understanding of the current policy landscape and whether actions are appropriately targeted has potential to provide insight on whether

resources are being appropriately allocated. The aim of this study was to use a behavioural science approach to identify and categorise the policy levers being used in England that have potential to influence obesity across the early life course (here considered to be pregnancy and 0–5 years), identify how these interface with energy balance behaviours (behaviours related to energy intake or expenditure), and identify areas where further policy actions could focus.

## Methods

The objectives were to: 1) identify and describe the key factors relevant to obesity prevention in the early years; 2) identify and describe relevant English national policy; 3) systematically map the identified policies onto the key factors relevant to obesity prevention; and 4) identify gaps and where further policy actions could be most effectively focused.

A behavioural science approach was taken using the Capability-Opportunity-Motivation-Behaviour (COM-B) model and Behaviour Change Wheel (BCW) framework; see Fig 1 and Table 1 [20, 21]. The COM-B model identifies the factors required to bring about behaviour change and the BCW highlights the approaches likely to be effective. For a behaviour to occur, there must be sufficient Capability (physical and psychological); Opportunity (physical and social); and Motivation (reflective and automatic). COM-B and the BCW allowed us to theoretically describe obesity risk factors and existing policies, and then to create a map of how policies interface with risk factors. The methods used to address each aim are described below. Data extraction templates and detailed step-by-step coding guidelines were developed (provided in S1 File).

*To address objective 1 (identify and describe the key factors relevant to obesity prevention in the early years)*, the Foresight Systems map was used to identify key mechanisms influencing energy balance [22]. This map provides insight into the complex relationships between the many determinants of obesity, with factors organised into broad groups including those relating to an individual's physiology, physical activity, biology, diet, and psychology, the physical activity environment, and the food production system. The variables most strongly connected to 'energy balance' in the centre of the model were the focus of this study. These include 'key variables' and 'first tier variables', which are considered leverage points for obesity policy having strong connections with the wider system [23]. Only variables relevant to children were used, identified using the relevant Foresight map ("Segmented Map: Children") [22], including those occurring via parent actions; this resulted in 23 variables. Each variable was coded, by considering how it could influence energy balance behaviours using COM-B; one or more components of capability, opportunity or motivation were assigned. For example, the variable 'food exposure' was coded as 'physical opportunity' and the variable 'level of recreational activity' was coded as 'physical capability', 'physical opportunity', 'social opportunity', and 'reflective motivation'. Variables were independently coded by two researchers (HC and SR), the rate of agreement was recorded and discrepancies discussed to reach consensus.

*To address objective 2 (identify and describe relevant national policy)*, a systematic online search of current relevant national government policies was undertaken (see S1 File for search strategy). This included policy documents relating to obesity, child health (pregnancy and maternity care, child care settings, primary schools, leisure facilities), and other areas (dental; social e.g. travel, housing). Policy actions have been defined as 'the specific actions put into place by any level of government and associated agencies to achieve the public health objective' [24]. Therefore, we included interventions as well as policies providing they were judged to be 'concrete and actionable' (i.e. not aspirational or vague). Interventions and policies were systematically coded using the BCW. We considered how each intervention or policy could change behaviour, by one or more of the following intervention types: education, persuasion,

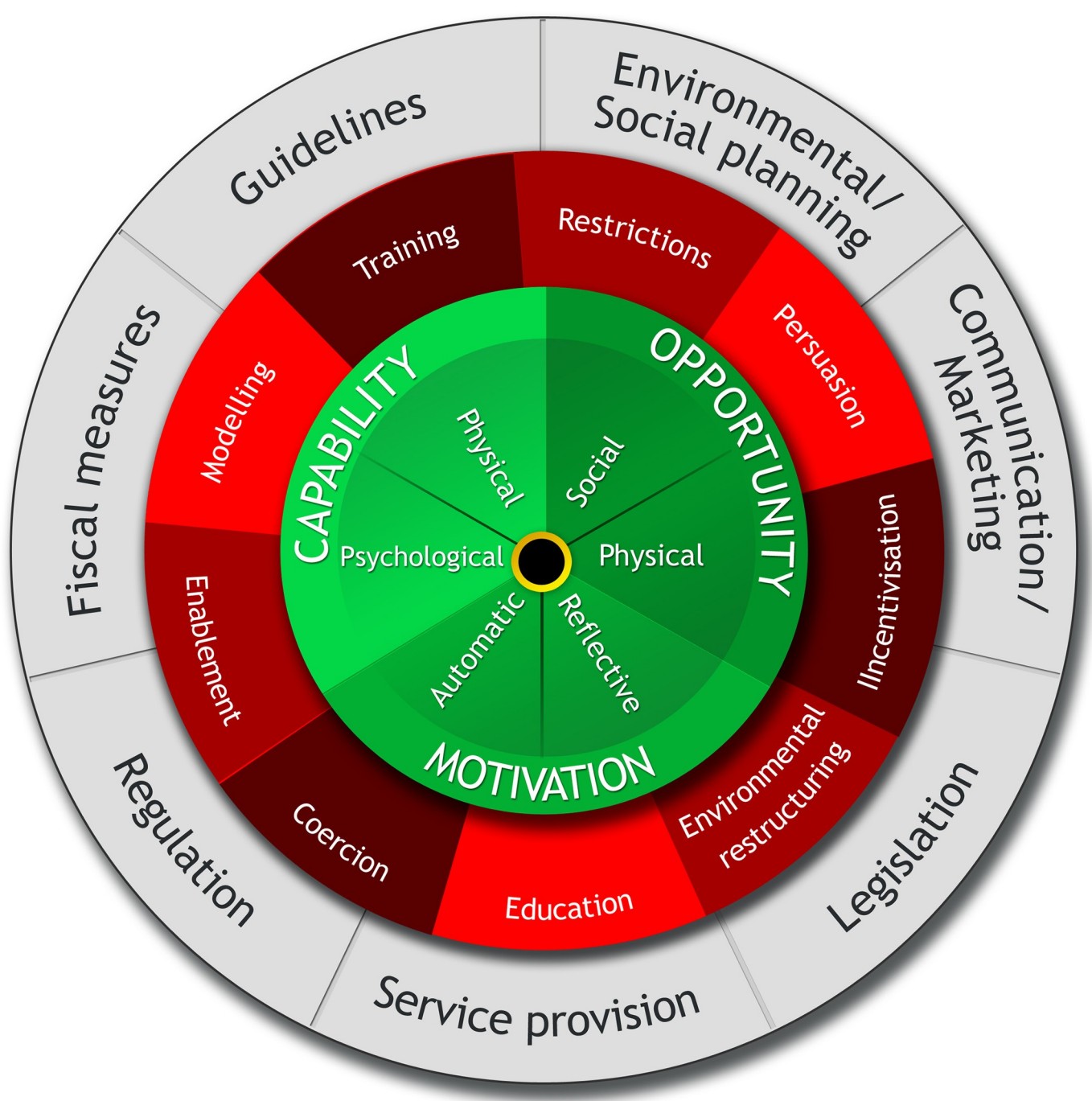

**Fig 1. Behaviour change wheel.** Reproduced with permission [21].

incentivisation, coercion, training, restriction, environmental restructuring, modelling, or enablement. We then considered which policy category or categories from the BCW (communication/ marketing, guidelines, fiscal measures, regulation, legislation, environmental/ social planning, service provision) best reflected the mode of action of each policy. Each policy was independently coded by two researchers (HC and AB), the rate of agreement recorded and any discrepancies discussed to reach consensus.

**Table 1. COM-B and Behaviour Change Wheel (BCW) components.**

| **COM-B components** | |
| --- | --- |
| Physical capability | Physical skill, strength, or stamina |
| Psychological capability | Knowledge or psychological skills, strength or stamina to engage in the necessary mental processes |
| Physical opportunity | Opportunity afforded by environment involving time, resources, location, cues, physical 'affordance' |
| Social opportunity | Opportunity afforded by interpersonal influences, social cues and cultural norms that influence the way we think about things, e.g. the words and concepts that make up our language |
| Automatic motivation | Automatic processes involving emotional reactions, desires (wants and needs), impulses, inhibitions, drive states and reflex responses |
| Reflective motivation | Reflective processes involving plans (self-conscious intentions) and evaluations (beliefs about what is good and bad) |
| **Behaviour Change Wheel intervention types** | |
| Education | Increasing knowledge or understanding |
| Persuasion | Using communication to induce positive or negative feelings or stimulate action |
| Incentivisation | Creating an expectation of reward |
| Coercion | Creating an expectation of punishment or cost |
| Training | Imparting skills |
| Restriction | Using rules to reduce the opportunity to engage in the target behaviour (or to increase the target behaviour by reducing the opportunity to engage in competing behaviours) |
| Environmental restructuring | Changing the physical or social context |
| Modelling | Providing an example for people to aspire to or imitate |
| Enablement | Increasing means/reducing barriers to increase capability (beyond education and training) or opportunity (beyond environmental restructuring) |
| **Behaviour Change Wheel policy categories** | |
| Communication/ marketing | Using print, electronic, telephonic or broadcast media |
| Guidelines | Creating documents that recommend or mandate practice. This includes all changes to service provision |
| Fiscal measures | Using the tax system to reduce or increase the financial cost |
| Regulation | Establishing rules or principles of behaviour or practice |
| Legislation | Making or changing laws |
| Environmental/social | Designing and/or controlling the physical or social environment |
| Service provision | Delivering a service |

Taken from [21].

The searches were undertaken in May 2018. The Childhood Obesity Plan: a call for action, Chapter 2 (COP2) was published in June 2018 and coded by HC and AG as a consensus exercise at a later date [25]. Policies were also coded according to the mode of influence; direct action on the child (e.g. change in food provided to the child in a child care setting), direct action via parent (e.g. policy resulting in parent changing food provided to child), indirect action via parent (e.g. policy resulting in parent changing own behaviour resulting in modelling of healthier behaviours), or a combination.

*To address objective 3 (systematically map the identified policies onto the key factors related to obesity prevention)*, the matrix developed by Michie et al (2014) was used, see Table 2 [21]. This indicates the intervention types most likely to be useful for bringing about change in each COM-B construct. The coding generated for objectives 1 and 2 was used. We examined the Foresight variables sequentially; we used the two sets of coding (COM-B and

**Table 2. Matrix of links between COM-B and intervention types.**

| COM-B components | Intervention types | | | | | | | | |
|---|---|---|---|---|---|---|---|---|---|
| | Education | Persuasion | Incentivisation | Coercion | Training | Restriction | Environmental restructuring | Modelling | Enablement |
| Physical capability | | | | | X | | | | X |
| Psychological capability | X | | | | X | | | | X |
| Physical opportunity | | | | | X | X | X | | X |
| Social opportunity | | | | | | X | X | X | X |
| Automatic motivation | | X | X | X | X | | X | X | X |
| Reflective motivation | X | X | X | X | | | | | |

Taken from [21]

X indicates the intervention types most applicable for bringing about change in each COM-B construct

BCW) to create a grid which we used to cross-reference the policies and intervention types. This enabled us to identify the specific policy actions available to address each Foresight variable. This was done by two researchers (HC and AB) as a consensus exercise; mapping of COP2 was additionally undertaken by HC and AG. The results from the mapping work were summarised into a HEAT map; green indicating the presence of at least one policy action to target a particular Foresight variable (with results presented for each relevant intervention type for each Foresight variable) and red indicating no identified policy actions.

*To address objective 4 (identify potential gaps and opportunities for policy actions to be developed or implemented)*, the mapping work generated for objective 3 was used. Coherence between the key influences of energy balance and current policies was examined (i.e. presence of policies to address key factors influencing energy balance). We also examined whether there were opportunities for developing or implementing policy. This is where there are factors known to influence obesity but for which we were unable to identify any policies to address, or where the policies were not targeted in the most effective way according to the BCW.

## Results

The coding of the Foresight variables using the COM-B model resulted in an agreement between coders of 71%, considered moderate, and agreement for the coding of policies using the BCW was 90%, considered very high [26]. The full coding is provided in S1 File.

### Objective 1: Identify and describe the key factors relevant to obesity prevention in the early years

The codes relating to COM-B assigned to the Foresight variables varied widely, so that there was a good spread of potential influences on children's energy balance; summarised in Fig 2.

**Intervention types.** Tr-training; En-enablement; Ed-education; Re-restriction; Env-environmental restructuring; Mo-modelling; Per-persuasion; Inc-incentivisation; Co-coercion; *Mapping*: Grey- not applicable (COM-B construct not relevant for that variable); Green- policy actions identified; Red- no policy actions identified, Orange -uncertain.

### Objective 2: Identify and describe national policy in the early years for obesity prevention

A total of 106 specific policy actions were identified in the initial search, this included specific policy initiatives (e.g. the 'National Child Measurement Programme' and 'Change for Life') and national clinical and public health guidelines produced by the National Institute for

| COM-B component | Physical capability | | Psychological capability | | | Physical opportunity | | | | Social opportunity | | | | Automatic motivation | | | | | | | Reflective motivation | | | |
|---|---|---|---|---|---|---|---|---|---|---|---|---|---|---|---|---|---|---|---|---|---|---|---|---|
| Foresight Variable \ Intervention type | Tr | En | Ed | Tr | En | Tr | Re | Env | En | Re | Env | Mo | En | Per | Inc | Co | Tr | Env | Mo | En | Ed | Per | Inc | Co |
| Stress | | | | | | | | | | | | | | | | | | | | | | | | |
| Food advertising | | | | | | | | | | | | | | | | | | | | | | | | |
| Food literacy | | | | | | | | | | | | | | | | | | | | | | | | |
| Self-esteem | | | | | | | | | | | | | | | | | | | | | | | | |
| Inconsistent messages | | | | | | | | | | | | | | | | | | | | | | | | |
| Degree physical edn | | | | | | | | | | | | | | | | | | | | | | | | |
| Food abundance | | | | | | | | | | | | | | | | | | | | | | | | |
| Food exposure | | | | | | | | | | | | | | | | | | | | | | | | |
| Innate activity | | | | | | | | | | | | | | | | | | | | | | | | |
| Recreation activity | | | | | | | | | | | | | | | | | | | | | | | | |
| Transport activity | | | | | | | | | | | | | | | | | | | | | | | | |
| Side effects drugs | | | | | | | | | | | | | | | | | | | | | | | | |
| Functional fitness | | | | | | | | | | | | | | | | | | | | | | | | |
| Palatability food | | | | | | | | | | | | | | | | | | | | | | | | |
| Breastfeeding/ weaning | | | | | | | | | | | | | | | | | | | | | | | | |
| Demand indulgence | | | | | | | | | | | | | | | | | | | | | | | | |
| Tendency to graze | | | | | | | | | | | | | | | | | | | | | | | | |
| De-skilling | | | | | | | | | | | | | | | | | | | | | | | | |
| Convenience of food | | | | | | | | | | | | | | | | | | | | | | | | |
| Portion size | | | | | | | | | | | | | | | | | | | | | | | | |
| Purchasing power | | | | | | | | | | | | | | | | | | | | | | | | |
| Energy density food | | | | | | | | | | | | | | | | | | | | | | | | |
| Market price food | | | | | | | | | | | | | | | | | | | | | | | | |

**Fig 2. HEAT map showing foresight variables with mapped policy action.**

Health and Care Excellence (NICE). Four additional policy actions were identified but not included as they either fed into other policies (update of Nutrient Profile Model), were planned for the future (Healthy Rating Scheme for primary schools; suite of digital applications for healthy eating) or were not a specific policy action (healthy marketing strategy). Some of the policies had more than one specific policy action (e.g. measurement and feedback elements of the 'National Child Measurement Programme' and a wide range of actions at different time points in the 'Healthy Child Programme'); the number of unique policies was 79. An additional nine policy actions were identified in COP2. In total, 115 specific policy actions were identified and 88 unique policies. The remainder of the results relate to these 115 policy actions. Examples of how these relate to the intervention types in the BCW, as well as the mode of action, is provided in Table 3. The most common intervention type was education, with environmental restructuring and modelling approaches also commonly used. Coercion, restriction and incentivisation approaches were rarely used. As well as policies impacting directly on the child, many were via adult-focused initiatives which provide educational opportunities for parents and modelling opportunities for children (via parent behaviour change). The number of policy actions assigned to each intervention type and policy category is shown in Figs 3 and 4 respectively.

## Objective 3: Systematically map the identified policies onto the key factors related to obesity prevention in early life

The results of the mapping work as a whole are shown in the HEAT map (Fig 2), which indicates where policy actions were identified for Foresight variable as per the BCW. Of a possible 287 opportunities for policy, over half had at least one policy action (n = 157; 54.7%), but we were unable to identify any actions for the remainder (n = 128; 44.6%) or there were uncertainties over the coding (n = 2; 0.7%). A summary of how well the identified policies mapped onto

**Table 3. Examples of how identified policy actions relate to BCW intervention types.**

| BCW intervention type | Examples of policy actions relating to each intervention type | | |
|---|---|---|---|
| | Direct action on child | Direct action via parent | Indirect action via parent |
| Education<br><br>*Increasing knowledge or understanding* | Food teaching in primary schools framework; School Food Plan; NICE guidance- obesity prevention- schools; resources to support NCMP in schools; Cycling and walking investment strategy (walk to school project) | Healthy Start Scheme; Healthy Child Programme; National Child Measurement Programme feedback; Change4Life resources; 5aday logos | Healthy Child Programme; One You NHS campaign |
| Persuasion<br><br>*Using communication to induce positive or negative feelings or stimulate action* | None identified | Promotion of breastfeeding within Healthy Child Programme; Clearer food labelling; consistent calorie labelling; Change4Life resources; NICE physical activity guidelines | One You campaign online resources; physical activity infographic |
| Incentivisation<br><br>*Creating an expectation of reward* | Cycling and walking investment strategy (walk to school project- children's challenge); | Change4Life vouchers for healthy food | None identified |
| Coercion<br><br>*Creating an expectation of punishment or cost* | None identified | None identified | None identified |
| Training<br><br>*Imparting skills* | Funding for bikeability training; Cooking in the national curriculum; Sports skills within national curriculum | NICE guidelines including imparting skills regarding breastfeeding, cooking, positive parenting | NICE guideline on obesity prevention (NHS interventions for adults) |
| Restriction<br><br>*Using rules to reduce the opportunity to engage in the target behaviour (or to increase the target behaviour by reducing the opportunity to engage in competing behaviours* | Advertising restrictions in children's media | Ban on sales of energy drinks to children; proposed ban of promotion of unhealthy foods and drinks | None identified |
| Environmental restructuring<br><br>*Changing the physical or social context* | Sugar drinks levy; Sugar and Calorie Reduction Programmes; provision of opportunities to be active at school (breaktimes and PE); Primary PE and Sports Premium; provision of healthier foods in school; Food in Early Years Settings; lowered maximum protein content of formula milk; Nursery Milk Scheme; universal free school meals for KS1; Daily mile | Sporting Future (improved local leisure facilities); local transport plans to encourage active travel; NICE guidance on postnatal care and maternal and child nutrition; consistent calorie labelling | NICE obesity prevention guidelines; NICE physical activity guidelines on walking and cycling |
| Modelling<br><br>*Providing an example for people to aspire to or imitate* | Creation of healthy food environments in public sector settings (including leisure centres); staff modelling recommended in Food in Early Years Settings; School Fruit and Vegetable Scheme; | Healthy Child Programme (peer support for breastfeeding); positive images within Change4Life; Baby Friendly accreditation | One You Campaign, NHS weight loss plan, various NICE guidelines |
| Enablement<br><br>*Increasing means/reducing barriers to increase capability (beyond education and training) or opportunity (beyond environmental restructuring)* | Universal free school meals for KS1; What works in schools and colleges to increase physical activity? | Vouchers within Healthy Start Scheme; behavioural elements and vouchers with Change4Life; prompts for behaviour change with Start4Life; | Support for behaviour change and access to resources in NICE guidelines; NHS weight loss plan |

the COM-B theoretical model and the intervention types from the BCW is shown in Table 4. For the COM-B constructs, 'physical capability' was the best covered and 'physical opportunity' and 'social opportunity' were reasonably well covered. The constructs with the least identified policy actions were 'psychological capability', 'automatic motivation' and 'reflective motivation'. For the intervention types from the BCW, there was wide variation in coverage. There were many policy actions based on education and environmental restructuring, with more than 80% of the opportunities for these approaches having policies in place. There were a reasonable number of policy actions in place to address modelling and enablement (with

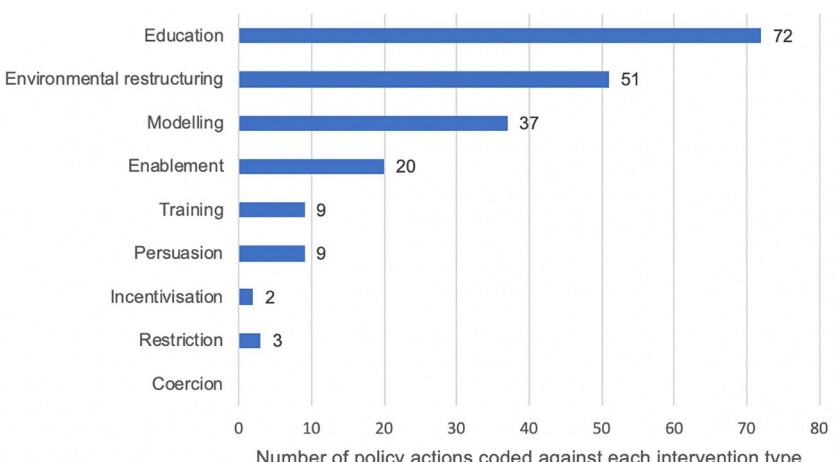

**Fig 3. Policy actions coded by 'intervention type'.**

over 60% of opportunities having policies in place), fewer for training, restriction, persuasion and incentivisation (less than 60% of opportunities) and no policy actions based on coercion.

The HEAT map (Fig 2) can also be used to examine the policy actions available for individual Foresight variables. The nutrition-related variables with the most policy actions in place were food literacy, food abundance, food exposure, palatability of food, breastfeeding and weaning, energy density (at least 60% of opportunities had identified policies). For example, actions for food literacy included policies for clearer food labelling and education within the Healthy Child Programme. There were a reasonable number of policies (40–60% of opportunities had identified policies) to address de-skilling, convenience of food, portion size, purchasing power, perceived inconsistency of science messages, and tendency to graze. For example, nutrient-based standards in food outlets to target food convenience. There is much overlap with many of the policy actions; for example, healthier foods in schools could help to address food abundance, food exposure, palatability of food offerings, and energy density. For physical activity, the variables with the most policy actions (at least 60%) were degree of physical education, innate activity in childhood, recreational activity, transport activity, functional fitness,

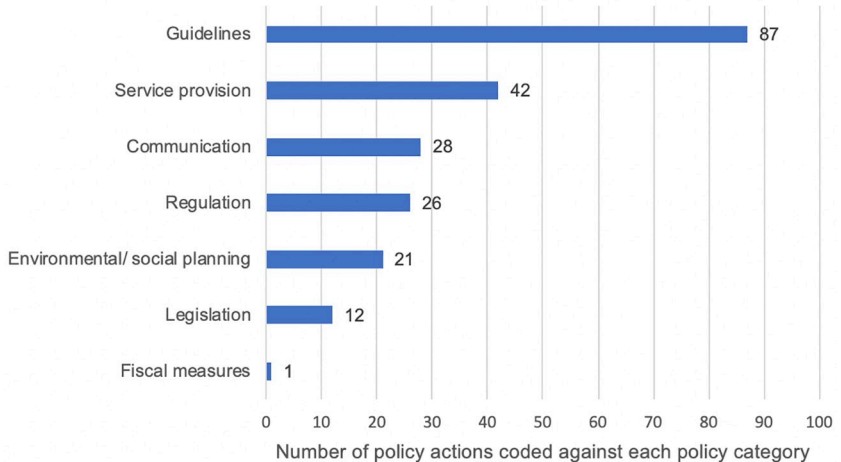

**Fig 4. Policy actions coded by 'policy category'.**

**Table 4. Policy actions in place by COM-B constructs and intervention types (across all foresight variables).**

| COM-B construct | Number with any mapped policies | % |
|---|---|---|
| Physical capability | 13/14 | 93% |
| *Physical skill, strength, strength or stamina* | | |
| Social opportunity | 42/60 | 70% |
| *Opportunity afforded by interpersonal influences, social cues and cultural norms that influence the way we think about things, e.g. the words and concepts that make up our language* | | |
| Physical opportunity | 38/60 | 63% |
| *Opportunity afforded by environment involving time, resources, location, cues, physical 'affordance'* | | |
| Psychological capability | 13/24* | 54% |
| *Knowledge or psychological skills, strength or stamina to engage in the necessary mental processes* | | |
| Reflective motivation | 22/52 | 42% |
| *Reflective processes involving plans (self-conscious intentions) and evaluations (beliefs about what is good and bad)* | | |
| Automatic motivation | 29/77* | 38% |
| *Automatic processes involving emotional reactions, desires (wants and needs), impulses, inhibitions, drive states and reflex responses* | | |
| **BCW intervention type** | | |
| Environmental restructuring | 37/41 | 90% |
| *Changing the physical or social context* | | |
| Education | 17/21 | 81% |
| *Increasing knowledge or understanding* | | |
| Modelling | 18/26 | 69% |
| *Providing an example for people to aspire to or imitate* | | |
| Enablement | 36/56 | 64% |
| *Increasing means/reducing barriers to increase capability (beyond education and training) or opportunity (beyond environmental restructuring)* | | |
| Persuasion | 13/24 | 54% |
| *Using communication to induce positive or negative feelings or stimulate action* | | |
| Restriction | 13/30 | 43% |
| *Using rules to reduce the opportunity to engage in the target behaviour (or to increase the target behaviour by reducing the opportunity to engage in competing behaviours* | | |
| Training | 17/41** | 41% |
| *Imparting skills* | | |
| Incentivisation | 6/24 | 25% |
| *Creating an expectation of reward* | | |
| Coercion | 0/24 | 0% |
| *Creating an expectation of punishment or cost* | | |

*There was uncertainty regarding the coding for one of the policy actions within each of these categories, these were therefore not coded as a mapped policy action

** There was uncertainty regarding the coding for two of the policy actions within each this category, these were therefore not coded as mapped policy actions

with a reasonable number (40–60%) present for de-skilling. For example, guidance for schools to target physical education and NICE guidance re public open spaces and schools to target recreational activity. The variables with limited identified policy actions were stress, food advertising, self-esteem, side-effects of drugs, demand for indulgence, and market price of

food (less than 30% of the opportunities for policies had identified policy actions). There were however some efforts to target these variables with marketing restrictions identified as key actions in the COP2.

### Objective 4: Identify potential gaps and opportunities for policy actions to be developed or implemented

As identified from the above mapping work, the COM-B constructs with the least identified policy actions were 'psychological capability', 'automatic motivation' and 'reflective motivation'. The HEAT map (Fig 2) indicates which intervention types specifically were lacking for these. In relation to increasing psychological capability, there were opportunities to strengthen policies through training and enablement. The Foresight variables that had gaps for these included stress, food literacy, inconsistent messaging, 'demand for indulgence', and de-skilling. In relation to increasing automatic motivation, there were, in particular, opportunities to strengthen policies based on persuasion, incentivisation, coercion, training, and enablement. In relation to reflective motivation, there were opportunities to particularly strengthen policies based on incentivisation and coercion. A number of Foresight variables had particular gaps relating to motivation, including stress, food advertising, self-esteem, 'demand for indulgence', tendency to graze, and market price of food. The Foresight variable 'market price of food' was notable in that no policies were identified which targeted it.

## Discussion

This scoping and mapping exercise identified where there were policies in place in England to target risk factors for obesity, and whether the methods employed were appropriate according to a behavioural science perspective, and where there was scope for additional actions. A substantial amount of policy activity was identified aiming to address childhood obesity and strong coverage of policies to target many of the energy balance-related risk factors from the Foresight systems map. A total of 115 relevant policy actions were identified and over half of the potential opportunities for addressing these risk factors had appropriate actions in place. This indicates that Government has implemented many actions in England to address early years obesity.

The mapping work in the current study was able to provide specific information about whether the policy action in place to target the Foresight variables used appropriate approaches, as identified by COM-B and the BCW. There were policy actions targeting all of aspects of the model but we also identified gaps. The COM-B model identifies two types of capability- physical and psychological. Physical capability refers to physical skill, strength or stamina and is best acted upon with training or enablement, whilst psychological capability refers to knowledge and psychological skills and is best acted upon with education, training and enablement. The majority of Foresight variables identified as being amenable to change through increasing physical capability related to physical activity (e.g. transport activity) and we identified good coverage of policy actions targeting these across most of the relevant Foresight variables. The majority of the Foresight variables related to psychological capability were psychological (e.g. stress, food literacy) or dietary (e.g. portion size) and we identified numerous policy actions targeting this based on education but gaps for actions based on training and enablement.

Within the COM-B model, opportunity comprises physical and social aspects. Physical opportunity relates to time, resources, locations and cues, whilst social opportunity relates to interpersonal influences, social cues and social norms. Both can be targeted using restriction, environmental restructuring and enablement; physical opportunity can also be increased with

training and social opportunity can be increased with modelling (i.e. setting a 'good' example). The majority of the Foresight variables across several domains (diet, physical activity, diet, economic) were related to opportunity and we identified reasonable coverage of policy actions across all of these. The Foresight variable 'market price of food' was a notable exception with no policy actions identified for either physical or social opportunity. The COM-B model goes onto identify reflective motivation (including self-conscious intentions and making evaluations) and automatic motivation (including emotional responses, impulses, and inhibitions). Both reflective and automatic motivation can be increased using multiple approaches from the BCW. The majority of the Foresight variables across several domains (diet, physical activity, diet, economic, physiology) were related to either reflective or automatic motivation. We identified numerous gaps in the coverage of policy actions to target these Foresight variables, with less than half having any actions. In particular, there were few policy actions relating to motivation for the following Foresight variables: stress, food advertising, self-esteem, demand for indulgence, tendency to graze, and market price of food.

Looking across the components in COM-B and the Foresight variables, the most common policy approach (as per the BCW) that we identified was education, along with a focus on guidelines targeting environmental restructuring and policies encouraging modelling opportunities (indirectly acting on the child via parent/carer behaviour change). We identified opportunities to further develop policy actions focused on enablement, persuasion, incentivisation, restriction, and coercion. For example; restriction is a possible approach for increasing physical and social opportunity, so policies based on restriction could be developed to promote physical activity (including both recreational and transport activity). Potential examples of this are restrictions on car use near schools to promote active school journeys or adding restrictions to tablets to limit their use to encourage active recreational time. Another example; since incentivisation is a possible approach for increasing reflective and automatic motivation, consideration could be given to developing policies based on incentivisation in relation to the Foresight variable portion size, an example could be incentivising purchasing of smaller packaged snack foods. Together, this highlights the focus on education and indicates that there are opportunities to build upon efforts for upstream change. In particular, there are numerous opportunities for further developing policies which act via increases psychological capability, reflective motivation and automatic motivation. Strengthening policies which increase the latter, such as people's desires, emotions and inhibitions may be particularly powerful as they go beyond a reliance on people's self-conscious 'choices'.

There were a many policies addressing environmental change, with regards to both the food and physical activity environment. However, despite the UK's good record of developing evidence-based policy guidelines, implementation of guidelines (especially public health policies) has often been poor [27]. A study of implementation, using the Food Environment Policy Index to map out and rate policies targeting childhood obesity in England, included a rating of implementation by experts [28]. Implementation was rated highest for monitoring (of obesity, risk factors, diet), nutrient declarations on labels, access to information, availability of dietary guidelines, school food standards, and population level targets. Implementation was rated lowest for food subsidies, planning policies to encourage fruit and vegetables, and systems-based approaches. This supports our findings that upstream policies are particularly challenging to implement. A focus on strengthening existing policy recommendations to facilitate implementation, especially those targeting upstream actions, may be useful. An example of such an upstream action is the price of food which can have a huge impact on people's purchasing decisions, with less healthy foods typically costing less and being consumed more by lower SES groups than healthier foods [29]; a lack of activity here may result in other policies having limited influence.

The emphasis on education and the limited action targeting automatic motivation indicates a reliance on policies primarily focusing on individual level change. A recent study found that, for addressing obesity, governments from developed countries tended to concentrate on policy levers addressing individual-level change rather than the environment, even in countries (such as England) with a strong policy focus on childhood obesity [19]. A recent systems-mapping exercise examined how local authorities in England address obesity using the 'Action Mapping Tool' [18]. Consistent with our findings, this work found that whilst only a small proportion of the causes of obesity were coded as 'individual lifestyle factors', nearly 60% of the actions around obesity targeted individuals. This suggests that an individual-orientated approach is a common theme throughout both national and local obesity policy. Interventions based only on individual choice have limitations. They require families to perceive change as important and be in a position to make such changes. This is likely to be challenging for many families but particularly difficult in families from lower SES backgrounds; this may act to further widen the health inequalities apparent with obesity [30]. A recent review indicated that all intervention types risk widening health inequalities but complex interventions which are targeted at multiple levels (systems, community, individual) and in multiple settings (school, health, population) appear to have less negative effects, and fiscal measures may even bring benefits [31]. Successive UK governments' policies highlight obesity as a serious problem; however there is a political tension between state and individual responsibility. Health choices are assumed to be the individual's responsibility to control even though the behaviours leading to excess weight gain are acknowledged to be greatly influenced by the environment [32]. Consistent with the findings in the current study, previous government policy documents have focused on information provision to change behaviour [33]. One example comes from an analysis of the pledges within the Public Health Responsibility Deal which found that most pledges focused on providing information for consumers, rather than structural changes (e.g. reformulation) [34]. Consumer views echo this, with analyses of online reactions to news stories about obesity policy finding either contradictory views around responsibility [35, 36] or that blame is attributed to the individual [37, 38]. This discourse is at odds with the evidence for the important role of environmental factors in contributing to obesity [39]. Of note, a greater emphasis on restricting of unhealthy food advertising was observed with the mapping of COP2 policy actions suggesting a move to more upstream action. The recently published policy paper 'Tackling obesity: empowering adults and children to live healthier lives' [40] was prompted by increasing evidence of a link between obesity and severity of COVID-19 symptoms [41]. A combination of individual level approaches (a weight management campaign and expansion of NHS obesity services) and environmental measures (such as legislation for calorie labelling and greater advertising restrictions) are outlined [40].

## Strengths and limitations

This study systematically identified national policies on childhood obesity using an authoritative system analysis of risk factors for obesity (Foresight) and took a behavioural science approach to first describe risk factors and policies, and then to conduct mapping work. We believe that this is the first time that a comprehensive mapping of obesity policies has been conducted using this approach. This allowed behavioural targets and policies to be systematically described in detail, allowing the identification of gaps and opportunities for further policy development. These gaps and opportunities were specifically characterised (according to the type of intervention, the method and the target) providing explicit information to inform the strengthening of current policy and future policy development. The work has potential to be built on and could be applied at a local level and used to inform needs assessments.

Our work is subject to a number of limitations. England is a populous country with high childhood obesity levels and a history of strong public health actions on obesity, thus findings are not necessarily generalisable to other countries. We focused on national-level policy and recognise that in most countries, including England, local or community policy actions may also be in place. We were unable here to include policy activity by the 152 upper-tier local authorities in England, but the approach used in the current study could be applied in that context. We included policy within primary schools, whilst this allowed us to capture actions directed at 4–5 year old children, this may have detracted from the early years focus. This analysis recorded actions and recommendations, not how well they were being implemented, which was beyond the scope of the study. Estimating the expected impact of policy actions on behaviour and weight was also beyond the scope of this work but could be useful since the number of policies available does not necessarily correspond to their expected impact. In particular, insight into the implementation of NICE guidelines would be useful, especially for public health guidelines where there may not be the same accountability as the clinical guidelines, which are included in service commissioning processes. Tools to support implementation include the Food EPI, an established method using expert consensus to provide policy ratings and identifies gaps and policy priorities [28] and surveillance plans [19]. The Foresight systems map was used to identify risk factors for childhood obesity as it is a comprehensive review of the evidence; however it was developed in 2007 so may not capture recent research. In addition, this is a fast moving field, the Childhood Obesity Plan (Chapter 3) was published as part of the Prevention Green Paper after the mapping work was completed and therefore not included [42], neither was the most recent policy paper addressing obesity [40]. There is also considerable interest in the role of the pre-conception period for later obesity risk [43]; however, the life-course stage for this work was restricted to pregnancy and early life to ensure that the study was feasible. The policy scoping was done via online searches, it is possible that there are additional policy actions not identified with this approach.

## Conclusions

This work generated a systematic map of current national government policy on obesity in England for the early years, with the purpose of identifying additional policy opportunities across the system. We identified a substantial amount of activity aiming to address early years obesity but scope for strengthening actions, especially upstream policies (acting on the environment and systems) and those targeting psychological factors contributing to obesity (stress, self-esteem, use of food for non-nutritive purposes). It is important to further consider implementation and likely impact of policy action. We found that using the COM-B model and BCW was useful for characterising the risk factors and existing policies, allowing for a detailed exploration of the current policy landscape and identification of gaps.

## Supporting information

**S1 File. Contains information relating to the searches and coding.**
(XLSX)

## Acknowledgments

Thank you to Heidi Yu Spurrell and Rosita Ibrahim who additionally assisted with coding.

## Author Contributions

**Conceptualization:** Corinna Hawkes, Russell M. Viner.

**Funding acquisition:** Corinna Hawkes, Russell M. Viner.

**Investigation:** Helen Croker, Simon J. Russell, Aswathikutty Gireesh, Aida Bonham.

**Methodology:** Helen Croker, Simon J. Russell, Corinna Hawkes, Helen Bedford, Susan Michie, Russell M. Viner.

**Project administration:** Helen Croker.

**Supervision:** Helen Croker.

**Visualization:** Helen Croker, Russell M. Viner.

**Writing – original draft:** Helen Croker.

**Writing – review & editing:** Helen Croker, Simon J. Russell, Aswathikutty Gireesh, Aida Bonham, Corinna Hawkes, Helen Bedford, Susan Michie, Russell M. Viner.

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
