## [Decision Letter · Decision Letter 0]

11 Aug 2020

PONE-D-20-19632

Obesity prevention in the early years: a mapping study of national policies in England from a behavioural science perspective

PLOS ONE

Dear Dr. Croker,

Thank you for submitting your manuscript to PLOS ONE. After careful consideration, we feel that it has merit but does not fully meet PLOS ONE’s publication criteria as it currently stands. Therefore, we invite you to submit a revised version of the manuscript that addresses the points raised during the review process.

Please see the reviewers comments and address them. I have given you a short deadline because comments are very minor. If you are on annual leave please let the editorial office know the deadline that you would need. 

We look forward to receiving your revised manuscript.

Kind regards,

Andrew Soundy

Academic Editor

PLOS ONE

Additional Editor Comments:

I agree with the reviewer so please address their points and return.

Journal Requirements:

Reviewers' comments:

Reviewer's Responses to Questions

**Comments to the Author**

1. Is the manuscript technically sound, and do the data support the conclusions?

Reviewer #1: Yes

2. Has the statistical analysis been performed appropriately and rigorously? 

Reviewer #1: Yes

3. Have the authors made all data underlying the findings in their manuscript fully available?

Reviewer #1: Yes

4. Is the manuscript presented in an intelligible fashion and written in standard English?

Reviewer #1: Yes

5. Review Comments to the Author

Reviewer #1: In this manuscript Croker H et al, have studied the national policies in England from a behavioural science perspective. This is a well designed approach and well presented.

Minor comment: The UK Government's new obesity strategy driven by the COVID-19 pandemic will be relevant to this review. A reference to this under discussion would add to this topic.

6. PLOS authors have the option to publish the peer review history of their article (what does this mean?). If published, this will include your full peer review and any attached files.

Reviewer #1: No

---

## [Author Response · Author response to Decision Letter 0]

19 Aug 2020

Please find enclosed revision of a mapping study of national policies in England for your consideration. We thank the reviewers for the very positive review and helpful comments.

We outline below the amends made in response to the following comment:

The UK Government's new obesity strategy driven by the COVID-19 pandemic will be relevant to this review. A reference to this under discussion would add to this topic.

We have added reference to this strategy in the discussion (page 21):

“The recently published policy paper ‘Tackling obesity: empowering adults and children to live healthier lives’ (40) was prompted by increasing evidence of a link between obesity and severity of COVID-19 symptoms (41). A combination of individual level approaches (a weight management campaign and expansion of NHS obesity services) and environmental measures (such as legislation for calorie labelling and greater advertising restrictions) are outlined (40).”

We have also added this to the limitations (page 22):

“In addition, this is a fast moving field, the Childhood Obesity Plan (Chapter 3) was published as part of the Prevention Green Paper after the mapping work was completed and therefore not included (42), neither was the most recent policy paper addressing obesity (40).”

We hope that this addresses this comment adequately. Thank you for your consideration

---

## [Decision Letter · Decision Letter 1]

7 Sep 2020

Obesity prevention in the early years: a mapping study of national policies in England from a behavioural science perspective

PONE-D-20-19632R1

Dear Dr. Croker,

We’re pleased to inform you that your manuscript has been judged scientifically suitable for publication and will be formally accepted for publication once it meets all outstanding technical requirements.

Kind regards,

Andrew Soundy

Academic Editor

PLOS ONE

Additional Editor Comments (optional):

Reviewers' comments:

Reviewer's Responses to Questions

**Comments to the Author**

1. If the authors have adequately addressed your comments raised in a previous round of review and you feel that this manuscript is now acceptable for publication, you may indicate that here to bypass the “Comments to the Author” section, enter your conflict of interest statement in the “Confidential to Editor” section, and submit your "Accept" recommendation.

Reviewer #1: All comments have been addressed

2. Is the manuscript technically sound, and do the data support the conclusions?

Reviewer #1: Yes

3. Has the statistical analysis been performed appropriately and rigorously? 

Reviewer #1: Yes

4. Have the authors made all data underlying the findings in their manuscript fully available?

Reviewer #1: Yes

5. Is the manuscript presented in an intelligible fashion and written in standard English?

Reviewer #1: Yes

6. Review Comments to the Author

Reviewer #1: (No Response)

7. PLOS authors have the option to publish the peer review history of their article (what does this mean?). If published, this will include your full peer review and any attached files.

Reviewer #1: No

---

## [Editor Report · Acceptance letter]

14 Sep 2020

PONE-D-20-19632R1

Obesity prevention in the early years: a mapping study of national policies in England from a behavioural science perspective

Dear Dr. Croker:

I'm pleased to inform you that your manuscript has been deemed suitable for publication in PLOS ONE. Congratulations! Your manuscript is now with our production department.

Kind regards,

on behalf of

Dr. Andrew Soundy 

Academic Editor

PLOS ONE